# CitH3, a Druggable Biomarker for Human Diseases Associated with Acute NETosis and Chronic Immune Dysfunction

**DOI:** 10.3390/pharmaceutics17070809

**Published:** 2025-06-23

**Authors:** Yuchen Chen, Zoe Ann Tetz, Xindi Zeng, Sophia Jihye Go, Wenlu Ouyang, Kyung Eun Lee, Tao Dong, Yongqing Li, Jianjie Ma

**Affiliations:** 1Division of Surgical Sciences, Department of Surgery, University of Virginia, Charlottesville, VA 22903, USA; dvj9dr@virginia.edu (Y.C.); asq4py@virginia.edu (Z.A.T.); cvk2pk@virginia.edu (X.Z.); ngs2fj@virginia.edu (S.J.G.); kl2ev@virginia.edu (K.E.L.); 2Human Biology DMP, University of Virginia, Charlottesville, VA 22903, USA; 3Department of Surgery, University of Michigan Health System, Ann Arbor, MI 48109, USA; ouyangwe@umich.edu (W.O.); taod@med.umich.edu (T.D.); yqli@med.umich.edu (Y.L.)

**Keywords:** sepsis, infection, inflammation, ischemia–reperfusion injury, wound healing, autoimmune diseases, antibody-based drugs

## Abstract

Neutrophils are essential components of innate immunity, executing a range of effector functions including phagocytosis, degranulation, and the release of neutrophil extracellular traps (NETs). A key hallmark of NET formation is the presence of citrullinated histone H3 (CitH3), produced by peptidylarginine deiminases (PAD2 and PAD4) to facilitate chromatin decondensation. While NETs play critical antimicrobial roles, excessive or dysregulated NET formation, termed NETosis, can drive tissue injury, chronic inflammation, and organ dysfunction across a wide spectrum of diseases. Beyond its structural role within NETs, CitH3 acts as a damage-associated molecular pattern (DAMP), amplifying immune activation and pathological inflammation. Elevated CitH3 levels have been identified as biomarkers in sepsis, viral infections, ischemia–reperfusion injury, organ transplantation, diabetic wounds, autoimmune diseases, and cancer. Despite increasing recognition of CitH3’s pathogenic contributions, its therapeutic potential remains largely untapped. This review summarizes recent advances in understanding the role of CitH3 in NETosis and immune dysfunction, highlights emerging strategies targeting CitH3 therapeutically, and identifies critical knowledge gaps. Collectively, these insights position CitH3 as a promising druggable biomarker for the diagnosis, prognosis, and treatment of acute and chronic inflammatory diseases.

## 1. Introduction

A neutrophil is a critical component of the innate immune system and acts as the initial frontline defender against invading pathogens [1,2]. Upon sterile or pathogen-induced infection or tissue injury, neutrophils are recruited from bloodstream circulation to the affected sites in response to a variety of signals, including pathogen-associated molecular patterns (PAMPs), damage-associated molecular patterns (DAMPs), cytokines, and chemokines [3,4,5]. These signals, along with the adhesion molecules they induce, are secreted or expressed by activated endothelial cells, resident immune cells, and injured epithelial cells [3,4,5]. At the sites of injury or infection, neutrophils are activated by pro-inflammatory stimuli, triggering a cascade of effector functions, mainly including phagocytosis, degranulation, the generation of reactive oxygen species (ROS), and the release neutrophil extracellular traps (NETs) [3,4,5,6].

NETs were first observed in 1996 by Takei et al. [7] and more thoroughly described by Brinkmann et al. in 2004 [8] as extracellular web-like structures built upon decondensed chromatin and coated with cytosolic and granule proteins [6,7,8]. The DNA in NETs is predominantly derived from the nucleus, with a minor portion that may originate from mitochondria [9]. The protein composition of NETs varies depending on the stimulus and usually include neutrophil elastase (NE), myeloperoxidase (MPO), and various antimicrobial peptides such as defensins and cathelicidins [6,10,11].

Among these, a particularly important and constantly observed marker is the citrullinated histone H3 (CitH3), generated by peptidylarginine deiminases (PADs) [6,12,13,14,15,16]. Primarily, the nuclear-localized PAD2 [12] and PAD4 [15] catalyze the hydrolytic conversion of arginine and methylarginine residues in histone H3 into citrulline, a post-translational modification that facilitates chromatin decondensation [6,12,13,14,15,16]. While NETs can neutralize and kill pathogens such as bacteria, viruses, and fungi, if dysregulated, the released CitH3 and DNA in NETs are implicated in causing damage to the host and contribute to pathogenesis of various diseases [17,18].

Due to its essential role in chromatin decondensation and the subsequent nuclear material released, CitH3 has emerged as a hallmark of NETosis—a process that involves excessive and dysregulated NETs [18,19]. Distinct from apoptosis and necrosis in morphology and mechanism, NETosis can occur via either lytic or non-lytic paths [20]. In the predominant lytic NETosis, activated neutrophils undergo depolarization, enabling nuclear envelope disassembly and chromatin decondensation into the cytoplasm. Subsequently, the plasma membrane ruptures, releasing NETs into the extracellular space, typically 3–8 h after activation [6,21,22]. By contrast, non-lytic NETosis involves the rapid secretion of chromatin and granule contents without compromising the integrity of the plasma membrane or nuclei, enabling the release of NETs within minutes upon exposure to stimuli such as *Staphylococcus aureus* [23,24]. Initially considered a protective mechanism against microbial infections, NETosis is now recognized as a key pathological driver of sepsis and other inflammatory diseases [25]. Excessive NETs formation is linked to increased susceptibility to infections and progressive organ dysfunction [26,27,28], and multiple studies have demonstrated the value of CitH3 in reflecting inflammation severity [28,29,30,31].

Beyond its structural role in NETs, CitH3 acts as a DAMP, capable of activating immune receptors and amplifying inflammatory signaling pathways upon its release into the extracellular space [32]. Upon release, CitH3 interacts with pattern recognition receptors (PRRs) such as TLR2 and TLR4 on macrophages, dendritic cells, endothelial cells, and epithelial cells, thereby triggering activation of the inflammatory pathways. This leads to the robust production of pro-inflammatory cytokines, including TNF-α, IL-6, and IL-1β, and further recruitment of neutrophils, amplifying tissue inflammation. Additionally, extracellular CitH3 contributes to endothelial dysfunction by promoting vascular permeability, cytotoxicity, and microvascular thrombosis, facilitating organ damage in severe inflammatory states such as sepsis. These features establish CitH3 not only as a biomarker of neutrophil activation but also as a direct mediator of innate immune overactivation and tissue injury. Accumulating evidence have suggested a pivotal role of CitH3 in immune cell death and inflammation in severe infections [13,33,34,35]. The improper clearance of lytic neutrophils and their remnants consistently triggers the innate immune system, leading to a cycle of uncontrolled, chronic inflammation and tissue damage.

As many diseases are linked to tissue damage and chronic inflammation, CitH3 serves as a vital biomarker for identifying both the diagnosis and prognosis of inflammation-related pathologies. Thus far, studies have shown that elevated CitH3 levels are present as a biomarker for both acute and chronic inflammatory-related diseases including (1) endotoxic septic shock [13,17,29,30,34,36,37,38,39,40,41,42,43,44,45,46,47,48,49], (2) viral infections [50,51,52,53], (3) ischemia–reperfusion injury [54,55,56], (4) organ transplantation [57,58], (5) diabetic wound [59,60,61,62,63], (6) autoimmune disease [9,64,65,66,67,68], and (7) cancer [27,69,70,71].

Despite growing interest in NET biology, a comprehensive understanding of the functional role of CitH3 in disease pathogenesis, as well as validation of CitH3 as a biomarker or therapeutic target, remains limited. In this review, we aim to provide an updated summary of current advances in understanding the role of CitH3 in diseases related to NETosis and acute/chronic inflammation. We highlight recent insights into emerging therapeutic strategies targeting CitH3 or excessive NETosis. We also discuss key knowledge gaps and propose future directions for research to fully elucidate the clinical relevance of CitH3 in both acute and chronic inflammatory diseases.

## 2. Acute Inflammation

CitH3 plays a pivotal role in the initiation and amplification of acute inflammation. During unregulated NETosis, excessive accumulation of extracellular CitH3 can trigger a cascade of inflammatory responses. CitH3 activates PRRs like TLR4 on immune cells, promotes activation of nuclear factor kappa B (NF-κB) signaling pathways, and upregulates the production of pro-inflammatory cytokines such as interleukin-6 (IL-6) and tumor necrosis factor-alpha (TNF-α), thereby recruiting additional neutrophils and perpetuating the inflammatory cycle [2,72,73,74]. This can disrupt tissue barriers, damage multiple cell types including endothelial cells, epithelial, immune cells and neural cells, and contribute to multiorgan dysfunction (Figure 1) [73,74].

These acute CitH3-mediated inflammatory responses are observed in both non-sterile inflammation, such as endotoxic shock and viral infections, and sterile inflammation, including ischemia–reperfusion (I/R) injury and tissue transplantation. The following sections will summarize how CitH3 contributes to pathogenesis across these contexts.

### 2.1. Non-Sterile Inflammation

#### 2.1.1. Endotoxic Septic Shock (Sepsis)

Sepsis is a life-threatening medical emergency caused by a dysregulated host response to infection, leading to multi-organ failure and high mortality rates. Sepsis can lead to critical organ failure including hepatic dysfunction, acute kidney injury, and various forms of endothelial dysfunction [41], with pulmonary injury being the most prevalent and fatal complication. Patients who suffer sepsis-induced lung injury during their ICU stay have a heightened risk of developing or worsening organ failure within the first year post-discharge [39,48,49].

The early phase of sepsis is often characterized by systemic inflammatory response syndrome, which can be managed with supportive care. However, as sepsis progresses, an uncontrolled cytokine storm and widespread cellular injury drive multiple-organ dysfunction, often leading to death despite aggressive treatment. Each hour of delay in treatment increases sepsis mortality by 4–9% [39,48,49]. While early interventions, such as respiratory stabilization, fluid resuscitation, and prompt antibiotic treatment, improve outcomes [43,75,76,77], no existing therapy directly targets the underlying drivers of sepsis: cytokine storms and infection-induced cellular damage.

Among these drivers, CitH3 has emerged as both a biomarker and a pathogenic effector in sepsis. Over the past two decades, researchers from our group and others have established a strong correlation between CitH3, sepsis severity, and mortality [13,17,29,30,31,32,33,34,35,36,37,38,39,40,41,42,43,44,45,46,47,48,49,78]. Elevated serum CitH3 levels in septic patients suggest that CitH3 is not only a biomarker of disease severity but also a critical mediator of sepsis-induced organ damage [29,30,37,79]. According to analyses of serum samples of 160 critically ill patients with septic and non-septic shock, and healthy volunteers, levels of circulating CitH3 at enrollment were significantly increased in the septic shock patients compared to patients hospitalized with non-infectious shock (*p* < 0.0001), and positively correlated with PAD2 and PAD4 concentrations and Sequential Organ Failure Assessment Scores (r = 0.36, *p* < 0.0001) [30]. Moreover, in septic patients, serum CitH3 levels at 24 h and 48 h post-admission were significantly higher (*p* < 0.01 and *p* < 0.05, respectively) in the survivors than those who did not survive [30].

Circulating CitH3 perpetuates a vicious cycle of NETosis and pyroptosis in immune cells, driving an uncontrolled cytokine storm that leads to pulmonary injury [34,37]. Additionally, CitH3 also regulates microvascular inflammation by disrupting endothelial barrier integrity [74]. It can weaken cell–cell adheren junctions and increase vascular permeability, particularly during sepsis, further escalating inflammation [74]. Given its central role in amplifying immune dysfunction, targeting CitH3 represents a promising therapeutic strategy for mitigating sepsis-induced multi-organ failure.

#### 2.1.2. Viral Infection

Viral infections represent another context in which innate immune defense may contribute to host pathology. Although direct research on CitH3 in viral infections remain limited, elevated levels of CitH3 have been observed in infections induced by influenza and SARS-CoV-2, both of which are characterized by pulmonary involvement and significant inflammatory responses.

In severe COVID-19 cases, viral pneumonia can progress to acute respiratory failure. Elevated levels of NETs and CitH3 levels were detected in hospitalized COVID-19 patients [52,53]. A negative correlation was observed between CitH3 levels and oxygenation, suggesting a potential contribution of CitH3 to respiratory dysfunction. Furthermore, influenza infection is correlated to high extracellular CitH3 levels in mouse lungs [50]. CitH3 showed strong co-localization with MPO and DNA in severely damaged areas with disrupted alveolar–capillary surfaces and vasculature. Bronchoalveolar lavage fluids collected at both 3- and 6-days post-infection exhibited elevated CitH3 levels, reinforcing the role of excessive CitH3 release in viral infection-induced inflammation and tissue damage. Nevertheless, further research is needed to explore the predictive power of CitH3 in longitudinal cohorts and determine the extent to which CitH3 may be a novel therapeutic target in severe viral infection.

### 2.2. Sterile Inflammation

Sterile inflammation, triggered by non-infectious tissue damage, elicits robust innate immune responses in the absence of microbial pathogens. Elevated CitH3 levels are present in multiple forms of sterile inflammation such as brain ischemia, cardiac arrest, organ transplants, kidney injuries and excessive acetaminophen (APAP)-induced liver injuries [40,54,55,57,80,81]. Here, we focus on ischemia–reperfusion (I/R) injury and organ transplantation, where CitH3 has emerged as a pivotal mediator in amplifying inflammation and exacerbating tissue injuries.

#### 2.2.1. Ischemia–Reperfusion Injuries

Elevated levels of CitH3 have been detected in both experimental models and clinical patients with I/R injuries, correlating closely with the severity of organ damage. In APAP-induced acute liver injury mouse model, excessive APAP triggered multiple forms of programmed hepatocyte deaths, including pyroptosis, apoptosis, and necroptosis, accompanied by increased CitH3 levels in both liver tissue and serum [80]. In a rat model with permanent middle cerebral artery occlusion, elevated levels of circulating neutrophils and CitH3 in the cerebrospinal fluid were detected before they infiltrated the brain parenchyma. CitH3 levels increased in peripheral blood 12 h after stroke onset [56]. Moreover, in patients with acute ischemic stroke, elevated plasma levels of CitH3 at the onset were associated with all-cause mortality at one-year follow-up visits [56]. The early neutrophils may initially undergo non-lytic NETosis, transitioning into lytic NETosis when entering the brain, thereby amplifying NET formation and CitH3-mediated injury after infiltrating the brain parenchyma [56].

Recent studies also showcased CitH3’s predictive value in cardiac I/R injury and post-cardiac arrest outcomes. Circulating CitH3 levels were indicative of disease severity in patients after restoration of spontaneous circulation (ROSC) after cardiac arrest, and served as a predictor of 28-day all-cause mortality [54]. Blood collected from patients resuscitated from cardiac arrest, either survivors or non-survivors, showed elevated levels of circulating CitH3 compared to that of healthy volunteers. Moreover, CitH3 levels on days 1, 3, and 7 after ROSC were significantly increased (*p* < 0.05) for the non-survivors compared to the survivors [54]. Notably, CitH3’s prolonged detectability offered a clinical advantage because it permits repeated measurements to monitor disease progression and prognosis over time [54].

Beyond systemic I/R injury following cardiac arrest and ROSC, CitH3 also plays a role in localized myocardial I/R injury. Following myocardial infarction (MI), attracted by cellular debris and DAMPs generated by necrotic cells, neutrophils rapidly appear at the infarct within hours [82]. Even subtle alterations in neutrophil resident time or density at the infarct can profoundly affect myocardial structure, potentially leading to catastrophic consequences [83]. Meanwhile, CitH3 levels were elevated in the blood of ST-segment elevation myocardial infarction (STEMI) patients compared to healthy controls [84]. In this context, excessive CitH3 can exacerbate cardiac injury and adverse remodeling, while further investigation is still warranted to decipher the nature and the consequences of CitH3 elevation in MI models, identify the optimal window for therapeutic intervention, and design CitH3-based treatments to reduce the risk of heart failure.

#### 2.2.2. Organ Transplantation

Given that I/R injury is a critical component of many transplant procedures, particularly during graft retrieval and implantation, it is rational that NET formation may also play a role in transplantation outcomes. Although current research in this area remains limited, there is evidence suggesting that CitH3 could represent promising targets to improve transplant success.

In a cohort of 93 hospitalized patients who underwent orthotopic liver transplantation, elevated CitH3 levels were associated with in-hospital mortality [57]. Patients who developed graft rejection and infection had higher levels of CitH3 prior to transplantation [57]. Furthermore, CitH3 levels increased after graft reperfusion, then declined before discharge [57], corresponding with key phases of transplant physiology. Similarly, in the context of lung transplantation, where primary graft dysfunction (PGD) remains a leading cause of early mortality and can also contribute to long-term graft failure, NETs and CitH3 accumulate in the lung during both experimental and clinical PGD [58].

## 3. Chronic Immune Dysregulation

Elevated circulating levels of CitH3 contribute to both acute inflammation and chronic inflammation [2,22,66,68,85,86,87]. In chronic conditions such as diabetic wounds and autoimmune disorders, sustained immune activation drives persistent NET formation, maintaining elevated CitH3 concentrations. While acute inflammation typically causes transient spikes in CitH3, chronic diseases lead to prolonged elevations of CitH3.

Persistent inadequate clearance of CitH3 promotes polarization of pro-inflammatory M1 macrophage, perpetuating chronic inflammation (Figure 2), which is commonly seen in diabetes, autoimmune disease, and cancer [68,69,70,71,85,88,89]. This pathological process contrasts with normal wound healing, where anti-inflammatory M2 macrophages facilitate tissue repair and regeneration through the secretion of growth factors and cytokines [90]. Here, we summarize and discuss the role of CitH3 in chronic immune dysregulation, with particular focus on its implications in diabetic wounds, autoimmune diseases and cancer.

### 3.1. Diabetic Wounds

Wound healing is an intricate process that encompasses the actions of various cell types and signaling pathways. While acute wounds like surgical, traumatic, or burn injuries typically heal within weeks, chronic wounds such as diabetic foot ulcers (DFUs) often persist for months due to underlying pathological conditions like diabetic mellitus and bacterial infection [91,92]. Wound healing impairment in diabetes, particularly DFUs, poses significant risks of morbidity and mortality [93,94,95]. The combination of neuropathy and vasculopathy contributes to DFUs, yet the precise cellular and molecular mechanisms impairing tissue healing in diabetes remain poorly understood. This gap limits therapeutic interventions beyond conventional glucose control, revascularization, and standard wound care.

Delayed wound healing due to uncontrolled NETosis has been well-reported in patients with diabetes [61,96,97]. Excessive NET extrusion and inefficient macrophage clearance increases in situ inflammation and altered both fibroblast and keratinocyte pro-healing action [60]. Followed by wound injury, DAMPs and PAMPs are released from wound tissues [60,91]. The PRRs on the membranes of neutrophils are thus activated. Consequently, neutrophils are immediately recruited from the circulation to the wound site for suppressing progressive deterioration [6,63,91,98].

CitH3 levels were elevated in the wounds of diabetic mice compared to normoglycemic controls. Moreover, PAD4 knockout mice, which lack the ability to citrullinate histones, demonstrated accelerated wound healing under diabetic conditions [59]. These findings support CitH3 as a potential biomarker and also therapeutic target of NETosis in diabetic wounds. At present, no NETosis-targeted therapies have received regulatory approval for treating chronic diabetic wounds. Current anti-NETosis strategies such as DNase-1 and PAD inhibitors [14,51] rely on global inhibition of NET-associated pathway and are associated with off-target effects due to their roles in other physiological processes. Therefore, their clinical translation remains limited. In this context, selectively targeting CitH3 may provide a more precise and clinically viable therapeutic approach.

Additionally, CitH3 may be a broader biomarker for diabetic inflammation. A recent study found that circulating CitH3 levels were higher in T2DM patients with glycated hemoglobin (HbA1c) values greater than 7.0% [88]. Circulating CitH3 also positively correlated with clot lysis time, suggesting a link with thrombosis risk associated with tissue damage and cell death in T2DM [88]. Whether CitH3 plays a central role in multiple diabetes-associated inflammatory comorbidities, and whether it holds promise as both a biomarker and therapeutic target, warrants further investigation.

### 3.2. Autoimmune Disease

CitH3-mediated M2-M1 shift contributes to chronic inflammation, particularly in autoimmune diseases where CitH3-containing NETs are abundant, such as rheumatoid arthritis (RA), inflammatory bowel disease (IBD), and systemic lupus erythematosus (SLE).

Patients with RA exhibit elevated levels of improperly activated circulating neutrophils, which exacerbate inflammation and joint damage through excessive NETosis, thereby contributing to disease progression [66]. According to a recent study investigating serum CitH3 levels of 151 RA patients with varying disease severity and 56 healthy controls, patients with highly and moderately active RA had higher CitH3 concentrations compared to those with mild disease or in remission, whose levels were still elevated compared to controls [99]. Notably, CitH3 was identified as an effective biomarker for distinguishing RA disease severity [99]. Additionally, in the majority of RA patients, pro-inflammatory cytokines and autoantibodies can induce rheumatoid arthritis-associated interstitial lung disease (RA-ILD). Neutrophils, and particularly the release of inflammatory NETs, are active in the progression of RA-ILD by maintaining interstitially inflamed and fibrotic conditions [100]. Circulating CitH3 was found to be elevated in RA-ILD patients compared to healthy controls, and positively correlated with interleukin-17A (IL-17A), a key cytokine in RA-ILD progression [100]. These together suggest CitH3 can serve as a biomarker for evaluating disease progression in RA and offer potential therapeutic opportunities to mitigate RA progression by neutralizing CitH3 molecules.

CitH3 also plays a role in IBD, given the presence of excessive NETs in patient’s colonic samples. In Crohn’s disease (CD), one of the primary IBD subtypes alongside ulcerative colitis (UC), excessive recruitment and accumulation of neutrophils contribute to key disease characteristics, such as mucosal injury and increased permeability to bacteria in the bowel tissues. CitH3 levels in intestinal tissues of CD patient were positively correlated with disease histopathological scores, which is similar to observations in UC patients [101,102]. Additionally, in patients with active IBD, CitH3 has been reported in biopsied colon tissue [103]. Consistently, in the colons in mouse colitis models, either induced by dextran sulfate sodium (DSS) or trinitrobenzene sulfonic acid (TNBS), CitH3 was elevated compared to the controls and co-localized with MPO and NE, contributing to impaired gut permeability and pathogenesis of mucosal inflammation [103,104]. Additionally, therapeutic anti-NETosis interventions, including PAD inhibitors, DNase I, and TNF-α inhibitors, have been shown to attenuate NET formation and protect mice against DSS-induced colitis, specifically by reducing intestinal inflammation, restoring intestinal integrity, and colitis-associated tumorigenesis [103,104]. These observations highlight the pathogenic role of CitH3 in IBD and indicate potential therapeutic opportunities by neutralizing the CitH3 molecule.

In SLE and its early stage, incomplete systemic lupus erythematosus (iSLE), CitH3 showed potential value as a biomarker of disease progression. iSLE patients exhibited higher levels of CitH3-DNA complexes compared with healthy volunteers [68], suggesting that dysregulated NETosis occur early during SLE development, with CitH3 serving as an indicator of this pathological process and offering potential utility as a biomarker for early diagnosis.

### 3.3. Cancer

Chronic inflammation is thought to activate malignant cells, although the mechanism of this remains unclear [105]. NETs have been identified in various cancer types including lung, pancreatic, colorectal, thyroid, and ovarian cancers, and are often suspected to promote cancer development [106,107,108,109,110,111,112]. A pan-cancer analysis of over 3000 solid tumors from 14 different cancer types revealed intratumoral neutrophils as the most adverse prognostic tumor-infiltrating leukocyte populations [113]. Mechanistically, NET-derived components such as HMGB1 or NE can bind to TLR9 or TLR4 to trigger mitochondrial biogenesis and stimulate cytokine production, establishing a feed-forward loop that sustains chronic inflammation and further NET formation. In addition, NETs contribute to tumor cell immune evasion by exhausting T cells or shielding tumor cells from T cell- or natural killer cell-mediated cytotoxicity [114,115,116]. Elevated serum CitH3 is associated with poor prognosis in in prostate cancer [117], hepatitis B virus-associated hepatocellular carcinoma [118], thyroid cancer [119], etc. However, the precise functions of circulating CitH3 in cancer and its association with anti-cancer treatments remain largely unclear at present.

In 2019, a study involving 957 patients with newly diagnosed or progressive cancer investigated CitH3 in association with arterial thromboembolism (ATE) risk and mortality [71]. Elevated initial CitH3 serum levels were associated with increased mortality, even after adjusting for age, sex, metastatic disease, and neutrophil count [71]. This association was pronounced in patients with lung cancer, lymphoma, and pancreatic cancer. Notably, among the biomarkers accessed, including cell-free DNA (cfDNA) and nucleosome concentrations, CitH3 was the only marker independently predictive of mortality, highlighting its clinical relevance [71].

Correspondingly, another set of patients with various advanced cancers exhibited a median CitH3 concentration approximately three times higher than that observed in both healthy controls and other severely ill non-cancer patients [70]. While CitH3 and cfDNA levels were positively correlated, possibly due to their co-release during NETosis, only elevated CitH3 was associated with a two-fold increase in the risk of short-term mortality [70]. Taken together, elevated circulating CitH3 levels potentially reflect cancer-associated processes and may be utilized as a biomarker or therapeutic targets for cancer patients, though further investigation is largely required.

Emerging studies highlight the crosstalk between NETs and tumor cells and different immune cells within the tumor microenvironment. Combining NET inhibitors (DNase or PAD inhibitors) with immune checkpoint blockers (anti-PD-1, anti-CTLA4) was found to enhance antitumor immunity, offering a rationale for NET-targeted combination therapies [120,121]. Whether the combination of CitH3 neutralizing antibodies with immunotherapy can further improve therapeutic outcomes remains an open and promising question.

## 4. Strategies for Targeting NETosis-Related Diseases

Targeting excessive NETosis is a potential therapeutic strategy for treating acute and chronic inflammatory disorders. There are several compounds referring to block or disrupt NETs, via either breaking downstream released inflammatory cytokines, DNA or NET-associated proteins, such as CitH3 antibody, DNase-1 and PAD2/4 inhibitors [14,17,34,51,122]. It is important to note that any therapeutic intervention must address safety concerns, e.g., such interventions cannot disrupt the innate immune response of the body.

### 4.1. DNase

One approach for combating the harmful effects of secondary necrosis following NETosis is to completely degrade already-formed NETs as a whole. In this context, DNases are utilized and demonstrated powerful effect in neutralizing and degrading NET-derived DNAs and reduce their destructive effects, thus alleviating excessive NETosis-induced inflammation and tissue damage [123]. The most commonly applied treatment is DNase I, with multiple studies demonstrating its power in reducing septic-induced acute respiratory distress syndrome and lung injury [124,125], and is approved for clinical use in the treatment of pulmonary cystic fibrosis [126]. However, some studies reported that DNase I treatment only showed little effects in reducing inflammatory cytokines and removing other NETs components like NE and CitH3 [127,128].

Considering the half-life of DNase I limits its activity, nanoparticles were adapted to load and deliver DNase I as a more consistently and stably approach and effectively suppressed SARS-CoV-2-induced cytokine storm [51,123]. Recently, a genetically engineered dual active DNase was successfully developed to realize more efficient NET degradation, by combining the activity of two distinct DNases DNase1 and DNase1L3 that preferentially digest double-stranded DNA and chromatin, respectively [129]. Nevertheless, certain pathogenic bacteria exploit extracellular DNases to evade destruction by the host immune system, indicating that while complete NET removal may alleviate NET-associated tissue damage, it also compromises the necessary protective antimicrobial functions of NETs [130]. The use of recombinant DNase can also lead to the release of cytotoxins from NETs and exaggerate inflammation or even autoimmunity, indicating the urgent need to explore alternative approaches to degrade NETs with the maximum benefits and minimum release of cytotoxic components [131,132].

### 4.2. PAD Inhibitors

Targeting the proteins involved in NET formation, specifically those regulating chromatin decondensation and release, is another widely applied approach. This includes inhibitors of PAD2 and PAD4, as well as MPO and NE. Early studies primarily focused on inhibiting PAD4 [15], as the importance of PAD4 in NET formation has been most well demonstrated. For examples, neutrophils form PAD4 knockout mice showed reduced citrullinated histones, and also failed to release NETs in response to LPS, H_2_O_2_, and bacterial infection [16,133]. Several PAD4 inhibitors, such as F-amidine, Cl-amidine, *o*-F-amidine, *o*-Cl-amidine and TDFA, have been developed to block PAD4-mediated signaling in both cancer and infection contexts [134,135].

Like PAD4, PAD2 can translocate from the cytosol into the nucleus to citrullinate histones, while PAD2-mediated citrullination of R26 likely facilitates transcriptional activation by opening the chromatin [12]. A newly developed PAD2 inhibitor, AFM41a promoted M2 macrophage polarization and autophagy in *Pseudomonas aeruginosa* (PA)-induced sepsis model, offering promising avenues for the treatment of PA infection and the improvement of sepsis outcomes [122]. Interestingly, it was reported that PAD2 inhibition specifically improved mice survival after LPS-induced endotoxic shock and decreased levels of circulating pro-inflammatory cytokines, whereas deficiency in PAD4 did not [17].

Nevertheless, PAD enzymes citrullinate a broad range of protein substrates without showing clear specificity [136], raising concerns that systemic PAD inhibition may result in unintended and unpredictable off-target effects. Therefore, more studies are warranted to map the global citrullination landscape following PAD2/PAD4 inhibition and to elucidate the functional consequences of PAD2/4-mediated citrullination on specific protein targets.

### 4.3. Other Approaches Targeting NETosis

Other approaches involve repurposing existing drugs to target activated neutrophils. An anti-malarial drug, hydroxychloroquine, was used for mitigating SLE, RA, and COVID-19 by preventing ROS and IL-8 production and TLR-9 expression, although it remains unclear if it can inhibit the NET formation-related functions of PADs, MPO, and NE [132]. Methotrexate was used to treat RA via inhibiting ROS production, and prednisolone was applied to treat autoimmune and inflammatory diseases in part by inhibiting neutrophil function [137,138,139]. Cyclosporine also showed potential inhibitory effect on NETosis, although with side effects that it may disrupt the overall immune system which causes an increase in recurrent infections [140].

Neutralizing monoclonal antibodies, including Rituximab, Belimumab, and Tocilizumab, have also been explored to inhibit NET formation by targeting B cells or inflammatory cytokines. Rituximab and Belimumab target CD20 and IgGγ, which can act in combination to reduce NET formation and deplete and inhibit autoreactive B cells. Tocilizumab targets the pro-inflammatory cytokine IL-6. Other potential targets of NETosis-neutralizing antibodies include NETosis downstream signals such as IL-1β and TNFα [44]. Currently, no NETosis-targeted therapies have been approved, and a summary of the above-mentioned strategies is provided in Table 1.

Overall, while preclinical studies have shown promise in targeting NETosis, clinical trials have been limited and often inconclusive [145]. A trial involving DNase I in COVID-19 (NCT04402970) showed modest benefit in improving oxygenation and decreasing DNA:MPO complex in BALF, with limitation to the time of drug delivery [146]. Efforts targeting PADs are mostly in preclinical stages. More robust clinical trials are warranted to validate the therapeutic potential of targeting NETosis. Current challenges include lack of specificity in NETosis-targeting agents, difficulty in measuring NETs reliably in vivo, and complex roles of NETs in both host defense and pathology.

In addition, multiple host-related factors, including lifestyle behaviors (e.g., smoking, alcohol use), demographic variables (age and sex), and metabolic conditions (obesity, and diabetes), can modulate NETosis by affecting neutrophil responsiveness and systemic inflammation [156,157,158]. Such variability may account for inter-individual and population-level differences in NET burden and disease progression, and should be taken into account in translational and clinical studies targeting NETosis.

It is worth noting that there are limitations associated with uncontrolled NET components inhibition, including inability to create functional NETs for pathogen infections or tissue damage when needed. Instead of inhibiting upstream factors for NETs, targeting downstream components could be an alternative strategy to alleviate global effect. Therefore, further investigations into its mechanisms and evaluation across diverse disease models are warranted.

### 4.4. Development of Antibody Against CitH3

A potential safe and effective means of harnessing excessive NETosis-induced inflammation and tissue injury is to capture CitH3 is with specific antibodies that recognize citrulline residues within their specific antigenic epitope. Although various commercialized antibodies are available, their efficacy has been found to be inconsistent [159]. Moreover, the commercially available CitH3 antibodies only recognizes histone H3 with citrullinated R2+R8+R17 [160], but H3 R26 is also citrullinated via PAD2 but not PAD4 [12].

Utilizing the CitH3 peptide with four citrulline residues at histone R2+R8+R17+R26, our group developed a new mouse anti-CitH3 monoclonal antibody, here referred to as mCitH3-mAb, which completely blocked the CitH3 catalyzed by both PAD2 and PAD4 and demonstrated a remarkable protective effect against endotoxic shock [34]. Deng et al. validated the target by finding that CitH3 increased human umbilical vascular epithelial cells permeability and induced NET formation. Compared to commercially available antibodies, this mCitH3-mAb exhibited a three-fold higher signal for the CitH3 protein. The study also showed that mCitH3-mAb improved survival compared to the commercial antibody following LPS-induced endotoxic shock, and protected against LPS-induced acute lung injury, as reflected by lower ALI score and decreased levels of L-1β and TNF-α in the lung homogenate [34].

To translate the findings on CitH3 into application for humans, our group has successfully developed a first-in-class humanized anti-CitH3 monoclonal antibody (hCitH3-mAb) using AI-based modeling and optimization of mCitH3-mAb. Evaluation on the safety and efficacy of hCitH3-mAb is ongoing. In addition, CIT-013, a recently developed monoclonal antibody with high affinity for citrullinated histones H2A and H4, has been shown to inhibit NETosis and reduce tissue NET burden a murine neutrophilic airway inflammation model, with anti-inflammatory consequences [64]. It remains to be explored whether the combination of CitH3 antibody with CIT-013 can exert synergistic effect in capturing citrullinated histones and suppressing excessive NETosis. Meanwhile, safety concerns should not be overlooked. Given that citrullinated histones from NETosis may also contribute to host defense regulations, a comprehensive assessment is warranted to evaluate the long-term impact of excessive or systemic blockade of CitH3 on innate immunity or tissue repair, especially in settings of chronic inflammation.

## 5. Future Perspectives, Challenges, and Conclusions

Numerous experimental strategies aimed at targeting neutrophil recruitment or activation demonstrated efficacy in limiting excessive inflammation or tissue injury. However, translating these strategies to the clinical arena has largely been unsuccessful. Various factors contribute to this gap, including improper dosing, the presence of concurrent risk factors, and the difficulty in identifying an optimal the therapeutic window. A major barrier to developing effective therapies lies in our incomplete understanding of the complex interplay between neutrophil-driven immune activation and host defense mechanisms. Currently, the complete inhibition of NETosis remains controversial. For example, while PAD4-deficient mice show reduced NETs and tissue damage in inflammatory models, some also display increased susceptibility to infections; although current anti-NETosis strategies such as DNase can reduce tissue damage associated with dysregulated NETosis, they also lead to the unintended release of toxic components, aggravating inflammation or autoimmunity. This underscores the urgent need for approaches that preserve the beneficial effect of NETs while mitigating the harmful effects of their dysregulation.

While NETosis is a systemic defense mechanism, tissue microenvironments significantly influence the dynamics and consequences of NET formation. For example, NETs have been extensively documented in pulmonary, renal, and vascular compartments, where abundant neutrophil recruitment occurs under inflammatory conditions. The lung, with its large capillary network and constant exposure to environmental stimuli, is a major site of NET-associated pathology, especially in ARDS, pneumonia, and COVID-19. In the kidney, glomerular NET deposition has been implicated in lupus nephritis [161]. Likewise, the rich vascularization may make it vulnerable to NETosis-induced injury. Indeed, an increased presence of NETosis was found in the ventricular myocardium in patients with heart failure due to cardiomyopathy [162]. The pathogenic effects of NETosis may be modulated by the local immune landscape and endothelial sensitivity. Thus, therapeutic interventions targeting NETs or CitH3 may need to be tailored to specific tissue environments to optimize efficacy and safety.

Single-cell transcriptomics have revealed functional heterogeneity among neutrophil subsets during inflammation [163], yet how CitH3 shapes this heterogeneity and consequently contributes to chronic inflammation remains unclear. Notably, no current single-cell studies have specifically addressed the regulatory role of CitH3 in immune cell programming, highlighting a critical gap in the field. Our group recently identified via scRNA-seq that PAD2/4 deficiency reduced Nlrp3^+^ macrophages and promote anti-inflammatory myeloid differentiation, suggesting citrullination may influence broader immune dynamics [164]. The single-cell approaches will serve as an important tool for future studies on how CitH3 shape the microenvironment during either acute or chronic inflammation.

CitH3 serves as both a biomarker and a circulating factor that induces downstream damage associated with NETosis when it escapes from the site of injury into the bloodstream. Circulating CitH3 activates PRRs on macrophages, epithelial and endothelial cells, promoting cytokine release, endothelial dysfunction, and microvascular thrombosis. These effects support its role as a direct contributor to tissue injury. This makes CitH3 a promising therapeutic target, distinguishing it from other common inflammatory targets such as IL-6, which are not specific to NETosis. Targeting CitH3 can prevent the propagation of inflammatory signals without impairing the formation of NETs or neutrophil activation. Thus, NETs can continue to function in pathogen defense and tissue repair without triggering the excessive inflammatory response and cytokine storm typically induced by CitH3 release. Importantly, the selective removal of excess CitH3 ameliorates dysregulated NETosis without compromising the body’s innate immune functions.

Meanwhile, standardizing CitH3 detection methods for clinical use remains a crucial need. Although techniques such as ELISA and immunofluorescence have shown feasibility, challenges remain in optimizing sensitivity and ensuring methodological standardization. To address these issues, Kurabayashi and colleagues have developed PEd-ELISA, a platform enabling rapid and sensitive CitH3 quantification [78,165]. Furthermore, deeper mechanistic studies are essential to fully elucidate the actions of CitH3 under physiological and pathological conditions, thus providing further insights into clinical diagnosis and therapeutic innovation.

In conclusion, antibody against CitH3 holds promise for bridging the gap between preserving innate immunity and mitigating NETosis-driven pathology, representing a clinically viable treatment option for dysregulated NETosis. The combination of highly selective anti-CitH3 antibodies with the sensitive PEd-ELISA platform provides a promising way to address current challenges including a lack of specificity in NETosis-targeting agents and difficulty in reliably measuring NETs in vivo. To fully realize its therapeutic potential, developing a humanized CitH3-mAb is crucial. Compared to earlier murine versions, a hCitH3-mAb has lower immunogenicity, enhanced human effector function engagement, and extended half-life in humans. Humanization of CitH3-mAb will represent a potential paradigm shift in bringing about a more effective and safe means of treating human diseases associated with dysregulated NETosis.

## Figures and Tables

**Figure 1 pharmaceutics-17-00809-f001:**
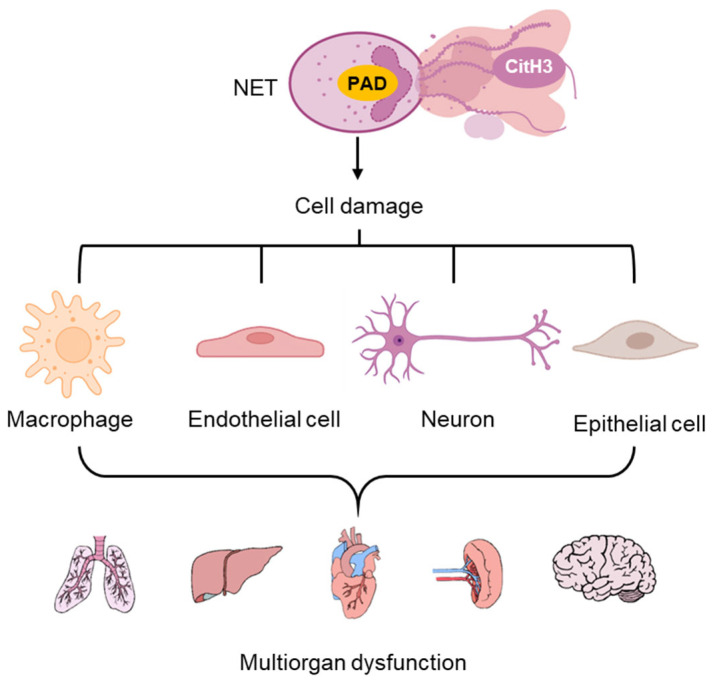
Cell damage and associated acute inflammation from NET-released CitH3. Elevated CitH3 release from NETosis caused injuries in different types of cells, which lead to multi organ dysfunction.

**Figure 2 pharmaceutics-17-00809-f002:**
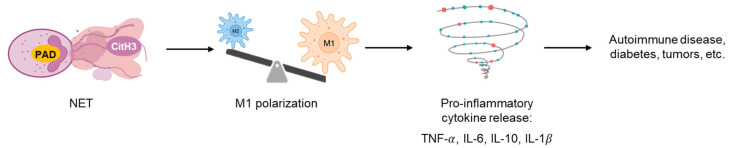
NETosis skews macrophage polarization toward the pro-inflammatory M1 phenotype. Sustained elevation of CitH3 and PAD-mediated protein citrullination disrupts the immunometabolic balance of macrophages, promoting a pro-inflammatory M1 state. This immune dysregulation contributes to the pathophysiology of autoimmune diseases, diabetes, cancer, and other chronic inflammatory conditions.

**Table 1 pharmaceutics-17-00809-t001:** Strategies for Targeting NETosis. Summary of current approaches taken to combat dysregulated NETosis to ameliorate inflammatory conditions.

Approach	Objective	Mechanism	Type of Study	References
Anti-CitH3 antibodies	Target the harmful CitH3 released during dysregulated NETosis	Capture circulating CitH3 released during NETosis, preventing excess inflammation and cytokine storm	Preclinical	[34,64]
Inhibitors against PAD2, PAD4, MPO, NE	Inhibit the function of proteins involved in NETs formation	Inhibit PAD2, PAD4, MPO, NE to suppress chromatin decondensation and nuclear material release	Preclinical	[65,132,141,142]
DNases I	Removal of NETs to alleviate harmful effects of NETosis	DNases degrade NETs	Phase 3; a more extensive clinical trial is warranted (NCT04402970)	[143,144,145,146]
Methotrexate, Prednisolone, Cyclosporine	Inhibit the function of activated neutrophils	Inhibit ROS and IL-8 production, TLR-9 expression. Indirectly inhibit NET production	Preclinical/Phase 3 with no results posted (NCT04227366)	[137,147,148,149,150,151,152]
Tocilizumab	Target inflammatory cytokines	Bind to IL-6 receptor to inhibit the activation of IL-6 downstream signaling pathways	Phase 2 (NCT04363736)	[153,154]
Rituximab and Belimumab	Target B cells	B cell depletion via inhibiting the survival and activation of B cells	Phase 2 (NCT02284984)	[155]

## Data Availability

No new data were created or analyzed in this study.

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
