# Peer review of "CitH3, a Druggable Biomarker for Human Diseases Associated with Acute NETosis and Chronic Immune Dysfunction"

_pharmaceutics, 2025, doi:10.3390/pharmaceutics17070809_

Round 1
Reviewer 1 Report
Comments and Suggestions for Authors
The manuscript titled "CitH3, a Druggable Biomarker for Human Diseases Associated with Acute NETosis and Chronic Immune Dysfunction" by authors
Y. Chen, Z. A. Tetz, X. Zeng, S. J. Go, W. Ouyang, K. E. Lee, T. Dong, Y. Li, and J. Ma submitted for publication in the journal Pharmaceutics is a well-written review that highlights citrullinated histone 3 (CitH3) as a promising biomarker in the diagnosis and potentially druggable target for a few debilitating inflammatory ailments. They highlight CitH3 as an underlying potential root cause for these diseases by providing enough supporting information. While the claims made in the manuscript needs further investigation, the authors have also developed a humanized CitH3 monoclonal antibody with extended half-life with enhanced effector function and reduced immunogenicity.
There are only minor errors such as the one on line 34, page 2 (danger-associated should be changed to damage-associated) and other grammatical/typos in the manuscript. Pending those corrections, the manuscript could be accepted for publication.
Author Response
Comments and Suggestions for Authors
The manuscript titled "CitH3, a Druggable Biomarker for Human Diseases Associated with Acute NETosis and Chronic Immune Dysfunction" by authors
Y. Chen, Z. A. Tetz, X. Zeng, S. J. Go, W. Ouyang, K. E. Lee, T. Dong, Y. Li, and J. Ma submitted for publication in the journal Pharmaceutics is a well-written review that highlights citrullinated histone 3 (CitH3) as a promising biomarker in the diagnosis and potentially druggable target for a few debilitating inflammatory ailments. They highlight CitH3 as an underlying potential root cause for these diseases by providing enough supporting information. While the claims made in the manuscript needs further investigation, the authors have also developed a humanized CitH3 monoclonal antibody with extended half-life with enhanced effector function and reduced immunogenicity.
There are only minor errors such as the one on line 34, page 2 (danger-associated should be changed to damage-associated) and other grammatical/typos in the manuscript. Pending those corrections, the manuscript could be accepted for publication.
Response: We appreciate the reviewer’s positive comments. We have corrected the indicated typo and carefully checked through the revised manuscript to avoid other grammatical/typos.

Reviewer 2 Report
Comments and Suggestions for Authors
Chen et al present a broad overview of the CitH3 as a druggable biomarker for human diseases associated with acute NETosis and Chronic Immune dysfunction. Authors have compiled an impressive work extensively describing the CitH3 and its role as a candidate biomarker, however before accepting, the manuscript can benefit from addressing a couple of major and minors’ comments.
Major
- Complement table 1 providing the clinical trial status (i.e. success status, etc),
- In order to reinforce the clinical relevance of the CitH3 authors should consider discussion the existing/future and successful/failed clinical trials. What is lucking, how it can be further pushed towards successful application in the clinics.
- Elaborate the discussion on the contradictory finding or negative results. For instance, do the high levels of CitH3 directly cause the pathology. Add the studies which can have balanced view on the CitH3 for the NETosis/state absence of them
- Provide more extensive mechanism how CitH3 acts as a DAMP
- Section 3.3. should be more elaborated, providing more insights into the types of cancer and possible mechanisms explaining it, as well current state on the filed
- Discuss what risk factors and other factors can affect the NETosis (i.e. smoking, drinking, sex, age)
- Add more tissue specific discussion on the NETosis
- Discuss safety concerns of completely blocking CitH3, elaborate the section 439-447
- Verify and make more specific the usage of the terminology “the significantly increase. Verify in the references if there is indeed a statistical test supporting this information (i.e. lines 135, 137, 159, 177, 192, 259, 282, 301, 325, 376, 431, 445, 483)
Minor
- Modify/complement Figure 1 and Figure 2, to address higher complexity of the immunological response. For instance, in Figure 1, what ‘cell damage’ cause multiorgan dysfunction? Do authors mention tissue immunity? Also provide studies references which organs are mentioned in figure 1
- Please add to discussion how novel single cell approaches are currently accessing the neutrophil heterogeneity and if there are any studies which address this mechanism of chronic inflammation regulation by the unregulated CitH3. If not, address this important knowledge gap in the filed
- For the sections of the manuscripts where authors mention the results obtained in their group, please provide references of other groups which perform similar work, or mention uniqueness of the studies otherwise. (i.e. 428/429/439)
Author Response
Comments and Suggestions for Authors
Chen et al present a broad overview of the CitH3 as a druggable biomarker for human diseases associated with acute NETosis and Chronic Immune dysfunction. Authors have compiled an impressive work extensively describing the CitH3 and its role as a candidate biomarker, however before accepting, the manuscript can benefit from addressing a couple of major and minors’ comments.
Major
- Complement table 1 providing the clinical trial status (i.e. success status, etc),
- In order to reinforce the clinical relevance of the CitH3 authors should consider discussion the existing/future and successful/failed clinical trials. What is lucking, how it can be further pushed towards successful application in the clinics.
Response: Thanks for the suggestion. As suggested, we have included clinical trial status in the revised Table 1 and added additional discussion on clinical trials (Line 445-458, 556-559), see text below:
“Overall, while preclinical studies have shown promise in targeting NETosis, clinical trials have been limited and often inconclusive[176]. A trial involving DNase I in COVID-19 (NCT04402970) showed modest benefit in improving oxygenation and decreasing DNA:MPO complex in BALF, with limitation to the time of drug delivery[177]. Efforts targeting PADs are mostly in preclinical stages. More robust clinical trials are warranted to validate the therapeutic potential of targeting NETosis. Current challenges include lack of specificity in NETosis-targeting agents, difficulty in measuring NETs reliably in vivo, and complex roles of NETs in both host defense and pathology.”
“In addition, multi host-related factors, including lifestyle behaviors (e.g., smoking, alcohol use), demographic variables (age, sex), and metabolic conditions (obesity, diabetes), can modulate NETosis by affecting neutrophil responsiveness and systemic inflammation [188–190]. Such variability may account for inter-individual and population-level differences in NET burden and disease progression, and should be taken into account in translational and clinical studies targeting NETosis.”
“The combination of highly selective anti-CitH3 antibodies with the sensitive PEd-ELISA platform provides a promising way to address current challenges include lack of specificity in NETosis-targeting agents and difficulty in reliable measurement of NETs in vivo.”
- Elaborate the discussion on the contradictory finding or negative results. For instance, do the high levels of CitH3 directly cause the pathology. Add the studies which can have balanced view on the CitH3 for the NETosis/state absence of them
Response: Thanks for the thoughtful suggestion. Accordingly, we have elaborated the direct link of CitH3 to pathology and added a more balanced view in the revised Discussion (Line 536-538, 504-509):
“Circulating CitH3 activates pattern recognition receptors (PRRs) on macrophages, epithelial and endothelial cells, causing injuries to these cells leading cytokine release, endothelial dysfunction, and microvascular thrombosis. These effects support its role as a direct contributor to tissue injury.”
“Currently, the complete inhibition of NETosis remains controversial. For examples, while PAD4-deficient mice show reduced NETs and tissue damage in inflammatory models, some also display increased susceptibility to infections; although current anti-NETosis strategies such as DNase can reduce tissue damage associated with dysregulated NETosis, they also lead to the unintended release of toxic components, aggravating inflammation or autoimmunity.”
- Provide more extensive mechanism how CitH3 acts as a DAMP
Response: As suggested, we have included more extensive description of the mechanism in the revised Introduction (line 74-83) as follows:
“Upon release, CitH3 interacts with pattern recognition receptors (PRRs) such as TLR2 and TLR4 on macrophages, dendritic cells, endothelial cells and epithelial cells, thereby triggering activation of the inflammatory pathways. This leads to robust production of proinflammatory cytokines, including TNF-α, IL-6, and IL-1β, and further recruitment of neutrophils, amplifying tissue inflammation. Additionally, extracellular CitH3 contributes to endothelial dysfunction by promoting vascular permeability, cytotoxicity, and microvascular thrombosis, facilitating organ damage in severe inflammatory states such as sepsis. These features establish CitH3 not only as a biomarker of neutrophil activation but also as a direct mediator of innate immune overactivation and tissue injury.”
- Section 3.3. should be more elaborated, providing more insights into the types of cancer and possible mechanisms explaining it, as well current state on the filed
Response: Thanks for the insightful suggestion. We have included further elaboration in the revised Section 3.3 (Line 331-344,361-367), as follows:
“Chronic inflammation is thought to activate malignant cells, although the mechanism remains unclear [131]. NETs have been identified in various cancer types including lung, pancreatic, colorectal, thyroid, and ovarian cancers, and are often suspected to promote cancer development [132–138]. A pan-cancer analysis of over 3000 solid tumors from 14 different cancer types revealed intratumoral neutrophils as the most adverse prognostic tumor-infiltrating leukocyte populations [139]. Mechanistically, NET-derived components such as HMGB1 or NE can bind to TLR9 or TLR4 to trigger mitochondrial biogenesis and stimulate cytokine production, establishing a feed-forward loop that sustains chronic inflammation and further NET formation. In addition, NETs contribute to tumor cell immune evasion by exhausting T cells or shielding tumor cells from T cell or natural killer cell-mediated cytotoxicity[140–142]. Elevated serum CitH3 is associated with poor prognosis in in prostate cancer[143], hepatitis B virus-associated hepatocellular carcinoma[144], thyroid cancer[145], etc. However, the precise functions of circulating CitH3 in cancer and its association with anti-cancer treatments remain largely unclear at present.”
“Emerging studies highlighted the crosstalk between NETs and tumor cells and different immune cells within the tumor microenvironment. Combining NET inhibitors (DNase or PAD inhibitors) with immune checkpoint blockers (anti-PD-1, anti-CTLA4) was found to enhance antitumor immunity, offering a rationale for NET-targeted combination therapies[146,147]. Whether combination of CitH3 neutralizing antibodies with immunotherapy can further improve therapeutic outcomes remains an open and promising question.”
- Discuss what risk factors and other factors can affect the NETosis (i.e. smoking, drinking, sex, age)
Response: We have included the discussion (Line 453-458) as follows:
“In addition, multi host-related factors, including lifestyle behaviors (e.g., smoking, alcohol use), demographic variables (age, sex), and metabolic conditions (obesity, diabetes), can modulate NETosis by affecting neutrophil responsiveness and systemic inflammation [188–190]. Such variability may account for inter-individual and population-level differences in NET burden and disease progression, and should be taken into account in translational and clinical studies targeting NETosis.”
- Add more tissue specific discussion on the NETosis
Response: We have included the discussion (Line 511-523) as suggested:
“While NETosis is a systemic defense mechanism, tissue microenvironments significantly influence the dynamics and consequences of NET formation. For example, NETs have been extensively documented in pulmonary, renal, and vascular compartments, where abundant neutrophil recruitment occurs under inflammatory conditions. The lung, with its large capillary network and constant exposure to environmental stimuli, is a major site of NET-associated pathology, especially in ARDS, pneumonia, and COVID-19. In the kidney, glomerular NET deposition has been implicated in lupus nephritis[193]. Likewise, the rich vascularization may make it vulnerable to NETosis-induced injury. Indeed, increased presence of NETosis was found in the ventricular myocardium in patients with heart failure due to cardiomyopathy [194]. The pathogenic effects of NETosis may be modulated by local immune landscape and endothelial sensitivity. Thus, therapeutic interventions targeting NETs or CitH3 may need to be tailored to specific tissue environments to optimize efficacy and safety.”
- Discuss safety concerns of completely blocking CitH3, elaborate the section 439-447
Response: Thank you for pointing out the need to discuss safety considerations. We have revised the relevant section to emphasize the importance of evaluating potential risks associated with systemic or prolonged CitH3 blockade, especially regarding its role in host defense and tissue homeostasis (Line 491-495).
“Meanwhile, safety concerns should not be overlooked. Given that citrullinated histones from NETosis may also contribute to host defense regulations, a comprehensive assessment is warranted to evaluate the long-term impact of excessive or systemic blockade of CitH3 on innate immunity or tissue repair, especially in settings of chronic inflammation.”
- Verify and make more specific the usage of the terminology “the significantly increase. Verify in the references if there is indeed a statistical test supporting this information (i.e. lines 135, 137, 159, 177, 192, 259, 282, 301, 325, 376, 431, 445, 483)
Response: Thank you for the valuable input. We have checked all the indicated studies, included additional key statistical data where applicable, and revised the text accordingly for those without a statistical test. Now these contents are more accurately presented.
Minor
- Modify/complement Figure 1 and Figure 2, to address higher complexity of the immunological response. For instance, in Figure 1, what ‘cell damage’ cause multiorgan dysfunction? Do authors mention tissue immunity? Also provide studies references which organs are mentioned in figure 1
Response: Thanks for the comments. Figure 1 and 2 have been revised accordingly with references cited.
- Please add to discussion how novel single cell approaches are currently accessing the neutrophil heterogeneity and if there are any studies which address this mechanism of chronic inflammation regulation by the unregulated CitH3. If not, address this important knowledge gap in the filed
Response: Thanks for providing this great vision. We have addressed this important knowledge gap in the revised Discussion (lines 524-532):
“Single-cell transcriptomics have revealed functional heterogeneity among neutrophil subsets during inflammation[195], yet how CitH3 shapes this heterogeneity and consequently contributes to chronic inflammation remains unclear. Notably, no current single-cell studies have specifically addressed the regulatory role of CitH3 in immune cell programming, highlighting a critical gap in the field. Our group recently identified via scRNA-seq that PAD2/4 deficiency reduced Nlrp3⁺ macrophages and promote anti-inflammatory myeloid differentiation, suggesting citrullination may influence broader immune dynamics[196]. The single-cell approaches will serve as an important tool for future studies on how CitH3 shape the microenvironment during either acute or chronic inflammation.”
- For the sections of the manuscripts where authors mention the results obtained in their group, please provide references of other groups which perform similar work, or mention uniqueness of the studies otherwise. (i.e. 428/429/439)
Response: Thanks for this helpful suggestion. The section describing our group’s development of hCitH3-mAb has been revised to emphasize its uniqueness as, to our knowledge, the first reported effort to humanize an antibody specifically targeting CitH3. We also provide context by referencing CIT-013, which targets other citrullinated histones (H2A and H4), to clarify how our approach differs and may offer complementary therapeutic value. (Line 484, 486-488).

Round 2
Reviewer 2 Report
Comments and Suggestions for Authors
Chen et al made significant improvements to the manuscript and very carefully addressed my major and minor comments, for what I am acknowledging them
I would recommend accepting this version of manuscript from Chen et al.